

# Strong genetic differentiation in tropical seagrass *Enhalus acoroides* (Hydrocharitaceae) at the Indo-Malay Archipelago revealed by microsatellite DNA

I Nyoman Giri Putra[1,2], Yuliana Fitri Syamsuni[3], Beginer Subhan[1], Made Pharmawati[4] and Hawis Madduppa[1,5]

[1] Department of Marine Science and Technology, Faculty Fisheries and Marine Sciences, Bogor Agricultural University (IPB), Bogor, Indonesia
[2] Department of Marine Science, Faculty of Marine Science and Fisheries, Udayana University, Bukit Jimbaran, Bali, Indonesia
[3] Indonesian Biodiversity Research Center, Denpasar, Indonesia
[4] Biology Department, Faculty of Mathematics and Natural Sciences, Udayana University, Bukit Jimbaran, Bali, Indonesia
[5] Center for Coastal and Marine Resources Studies, Bogor Agricultural University (IPB), Bogor, Indonesia

Corresponding author
Hawis Madduppa,
hawis@apps.ipb.ac.id,
hawis@ipb.ac.id

## ABSTRACT

The Indo-Malay Archipelago is regarded as a barrier that separates organisms of the Indian and Pacific Oceans. Previous studies of marine biota from this region have found a variety of biogeographic barriers, seemingly dependent on taxon and methodology. Several hypotheses, such as emergence of the Sunda Shelf and recent physical oceanography, have been proposed to account for the genetic structuring of marine organisms in this region. Here, we used six microsatellite loci to infer genetic diversity, population differentiation and phylogeographic patterns of *Enhalus acoroides* across the Indo-Malay Archipelago. Heterozygosities were consistently high, and significant isolation-by-distance, consistent with restricted gene flow, was observed. Both a neighbour joining tree based on $D_A$ distance and Bayesian clustering revealed three major clusters of *E. acoroides*. Our results indicate that phylogeographic patterns of *E. acoroides* have possibly been influenced by glaciation and deglaciation during the Pleistocene. Recent physical oceanography such as the South Java Current and the Seasonally Reversing Current may also play a role in shaping the genetic patterns of *E. acoroides*.

# INTRODUCTION

The Indo-Malay Archipelago is one of the most important land barriers (The Indo-Pacific Barrier, IPB), separating the Indian and Pacific Oceans (*Crandall et al., 2008*). Although the location of the exact boundary is debated (*Barber, Erdmann & Palumbi, 2006*; *DiBattista et al., 2012*), many species show strong genetic differentiation between the Indian and

Pacific populations, including marine plants, such as mangroves from the genus *Ceriops* (*Huang et al., 2008*) and the widely distributed *Bruguiera gymnorrhiza* (*Urashi et al., 2013*). Similar genetic differentiation between Indian and Pacific populations has also been demonstrated in diverse marine animals including reef fishes (*Thresher & Brothers, 1985*; *Gaither et al., 2009*), the marine gastropod, *Nerita albicilla* (*Crandall et al., 2008*), and the crown-of-thorns sea star *Acanthaster planci* (*Vogler et al., 2008*; *Yasuda et al., 2009*). Despite a growing number of such phylogeographic studies in the Indian and Pacific Ocean, only a few have tried to identify specific elements within the Indo-Malay Archipelago responsible for establishing and maintaining these barriers (*Carpenter et al., 2011*; *Keyse et al., 2014*).

The Indo-Malay Archipelago consists mostly of large land masses such as Sumatra, Thai-Malay Peninsula, Borneo, and the Greater Sundas. During the last glacial maxima (LGM), these islands coalesced to form a large land mass known as the Sunda shelf (*Voris, 2000*). Despite their similar geological origin, the population structure of flora and fauna in these islands is not homogenous. Studies of the mantis shrimp *Haptosquilla pulchella* have revealed a sharp genetic break across the Java Sea, which divides the population into north (Pacific) and south (Indian) (*Barber et al., 2002*). Other studies on the giant clam *Tridacna crocea* (*DeBoer et al., 2014*) and frigate tuna *Euthynnus affinis* (*Jackson et al., 2014*), showed that the populations of Sumatra in western Indonesia resembles that of the Indian populations, while Java resembles central Indonesian populations. Meanwhile, studies of the mangrove *Ceriops tagal* (*Liao, Havanond & Huang, 2006*) and the seagrass *Halophila ovalis* (*Nguyen et al., 2014*), revealed a concordance in the geographical barrier at the Thai Malay Peninsula, separating the Indian from the Pacific populations. Thus, the phylogeographic patterns in these regions are quite complex.

Gene flow could be driven by various factors, such as currents (*Barber, Erdmann & Palumbi, 2006*) and the geological history of a location (*Hart & Marko, 2010*). Geographical history such as the emergence of Sunda shelf during the Pleistocene period, is likely the main factor responsible for historically limited gene flow between the Indian and Pacific Ocean, which then, may have triggered lineage divergence in both oceans (*Carpenter et al., 2011*). In eastern Indonesia, the Halmahera Eddy and Indonesian Throughflow are likely candidates shaping biogeographic barriers between eastern and western Indonesia (*Barber, Erdmann & Palumbi, 2006*; *Carpenter et al., 2011*). In another example, water circulation and an eddy located at the southern tip of Sumatra plays a role in maintaining the genetic structure of mangrove *Rhizophora mucronata* Lam. in the Malay Peninsula and Sumatra (*Wee et al., 2014*).

Most phylogeographic studies across the Indo-Malay Archipelago to date have made use of mitochondrial genes from marine animals, such as crustaceans (*Barber, Erdmann & Palumbi, 2006*), reef fishes (*Nelson et al., 2000*; *Ackiss et al., 2013*), and bivalves (*DeBoer et al., 2008*). These studies have found concordant phylogeographic breaks between populations in the Indian Ocean and Java Sea. However, other marine animals, such as pelagic scads *Decapterus macrosoma* (*Arnaud-Haond, Bonhomme & Borsa, 1999*), and the marine gastropod *Nerita plicata* (*Crandall et al., 2008*), show no evidence of genetic structuring among these same regions. In contrast to the animals, study of marine plants, such as mangroves, with nuclear DNA markers revealed a genetic discontinuity of mangrove

*Rhizophora mucronata,* at the boundary between the Andaman Sea and Malacca Strait (*Wee et al., 2014*). Meanwhile, despite being an important foundational species, phylogeographic studies examining genetic patterns in seagrasses are currently lacking in this historically complex region.

Seagrasses are marine angiosperms that live in coastal areas on a substrate of sand, mud or a mixture of both, and most of their life cycle occurs below sea level. *Enhalus acoroides* is one seagrass species that is widely distributed in the Indo-Pacific from southern Japan, Southeast Asia, northern Australia, southern India and Sri Lanka (*Short & Waycott, 2010*). In Indonesia, *E. acoroides* can be found in Papua, North Maluku, Ambon, Sulawesi, Bali, Java, Borneo, and Sumatra (*Kiswara & Hutomo, 1985*). This species can be easily distinguished from other seagrasses because it has long leaves, the edges of the leaves are slightly rolled, and the rhizomes are thick and covered with black fibrous strands (which are the remnants of old leaves). Fruits of *E. acoroides* are capable of floating for up to 10.2 days (*Lacap et al., 2002*), during which they could reach a distance of between 0.1–63.5 km (*Lacap et al., 2002*). This might limit the dispersal of *E. acoroides*, although, occasional long distance dispersal (>1,000 km) of this species is also possible due to strong currents (*Nakajima et al., 2014*). Species with limited dispersal tendency are frequently hypothesized to be more genetically structured (*Bay, Crozier & Caley, 2006*) and thus, we expect to find significant population structure in this species.

Most seagrasses species are still categorized to be of least concern according to the IUCN criteria, but many researchers have reported that seagrasses populations are in continual decline around the globe (*Waycott et al., 2009*). Major loses of seagrass beds in Indonesia due to anthropogenic disturbances (for example, mining, coastal development, and polluted runoff) have been reported for decades, but restoration programs have only been initiated in the past few years (*Riani et al., 2012*). Population size reduction is known to cause loss of genetic diversity (*Allcock & Strugnell, 2012*) and a consequent elevated species extinction risk (*Spielman, Brook & Frankham, 2004*). In order to adequately preserve and manage seagrass ecosystems, the assessment of genetic diversity of seagrass species is essential.

Microsatellites are co-inherited and highly polymorphic markers (*Selkoe & Toonen, 2006*) and have been used broadly in population genetics and phylogeographic studies (*Poortvliet et al., 2013*; *Madduppa, Timm & Kochzius, 2014*; *Nakajima et al., 2014*; *Wee et al., 2014*). In this study, six previously developed microsatellite loci (*Nakajima et al., 2012*) were used to evaluate the genetic diversity, population structure and phylogeographic patterns of *E. acoroides* and to infer how the Sunda Shelf and regional currents shape the genetic patterns found in this species.

## MATERIALS AND METHODS

### Study area and sampling

A total of 202 *E. acoroides* samples from seven localities from Java and Sumatra were collected in 2014 (Fig. 1, Table 1). At each location, 18–42 individuals were taken in a zigzag pattern along a line transect. To avoid collection of the same genet or clone, only

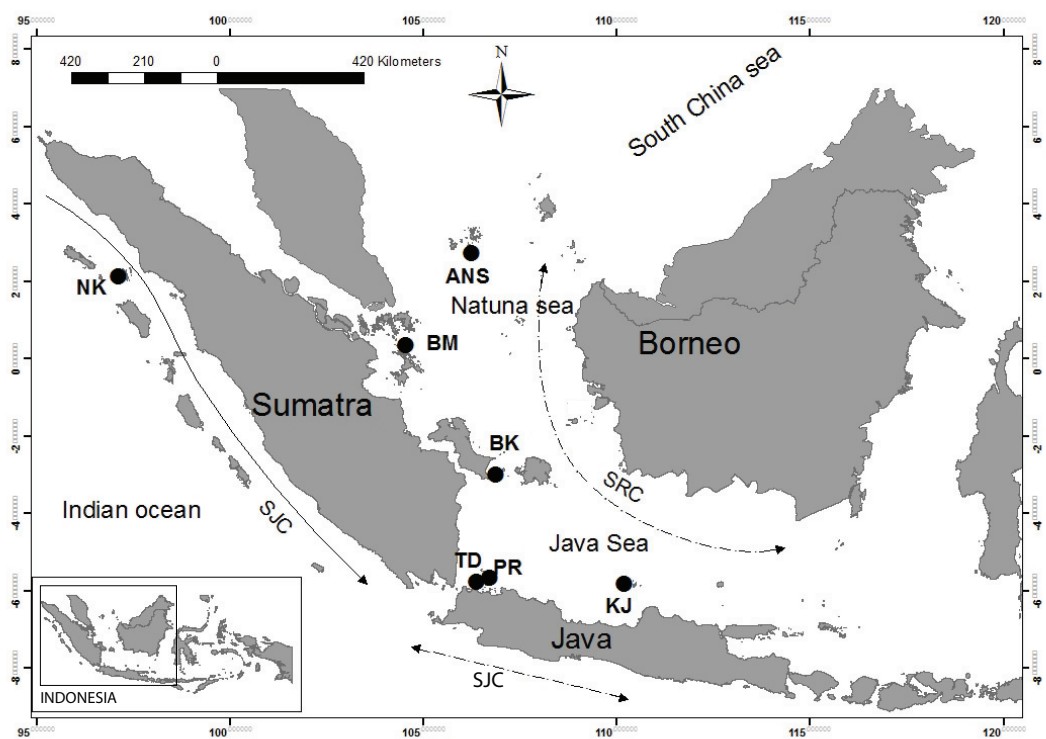

**Figure 1** **Sampling location from which *Enhalus acoroides* were collected for present study.** Sampling sites indicated by black circles. NK, Nakuri; BK, Bangka; BM, Batam; ANS, Anambas; TD, Tunda; PR, Pramuka; KJ, Karimun Jawa; SJC, South Java Current; SRC, Seasonality Reversing Current.

**Table 1** **Sampling location, abbreviation, geographical coordinates and the number of samples used in this study.**

| Collection site | Abbreviation | Latitude | Longitude | Number of individuals |
|---|---|---|---|---|
| Nakuri Island, Aceh | NK | 2.21° | 97.30° | 30 |
| Batam, Riau Archipelago | BM | 0.74° | 104.34° | 30 |
| Bangka island, Bangka Belitung | BK | −2.97° | 106.65° | 42 |
| Anambas, Riau Archipelago | ANS | 3.11° | 106.33° | 31 |
| Pramuka Island, Seribu Islands | PR | −5.74° | 106.61° | 18 |
| Tunda Island, Banten | TD | −5.81° | 106.28° | 27 |
| Karimun Jawa, Jawa Tengah | KJ | −5.86° | 110.40° | 24 |
| Total | | | | 202 |

one shoot was collected within a diameter of minimum of 5 m (DA Willette, pers. comm., 2013). Collected shoots were rinsed with fresh water to remove epiphytic algae. A young leaf from each shoot was desiccated with silica gel and preserved at room temperature until use.

**Table 2  Eight polymorphic microsatellite loci in *E. acoroides*: locus name, primer sequence, dye, product size range, repeat motif and GeneBank accession number.**

| Loci | Primer sequence (5′–3′) | Dye | Size range (bp) | Repeat motif | Accession no. |
|---|---|---|---|---|---|
| Eaco_001 | GGCTTGAGTTTGTTTAGAATTCTAG F<br>GGTTTTCCCAGTCACGACGTTACATGTGGAATGCATACAC R | FAM | 232–246 | $(TG)_{16}$ | AB689192 |
| Eaco_009 | CAATCGTCCAATCCAAAGGC F<br>GGTTTTCCCAGTCACGACGGGAGAATTGTATTATTTAC R | FAM | 142–154 | $(TG)_{13}$ | AB689194 |
| Eaco_019 | AGGTATTCCTTACCACCGTTC F<br>GGTTTTCCCAGTCACGACGCACGGAGGTCTTTCGAAGTTG R | VIC | 195–197 | $(CT)_7$ | AB689197 |
| Eaco_050 | GAATAAATCAAGTCCCTTGAG F<br>GGTTTTCCCAGTCACGACGCAAATAAGATGTGGCTTAC R | NED | 243–255 | $(TG)_9TA(TG)_5$ $TATG(TA)_8$ | AB689199 |
| Eaco_051 | CATACAGATGCATGCATACTC F<br>GGTTTTCCCAGTCACGACGCTAAGCGCTACGTGGTACTAG R | PET | 206–231 | $(GA)_{15}GTGC(GT)_{16}$ $GC(GT)_4$ | AB689200 |
| Eaco_052 | CAGGCGCACAACGTATGTAC F<br>GGTTTTCCCAGTCACGACGGAACCACATCATCAGTGTG R | NED | 147–149 | $(TG)_4TC(TG)_4$ $TC(TG)_5$ | AB689201 |
| Eaco_054 | GCTTCTAATTAGCATTTTGGACTTCAG F<br>GGTTTTCCCAGTCACGACGATTTGGGACGTCCAAAGAG R | PET | 267–295 | $(CT)_{15}$ | AB689202 |
| Eaco_055 | CTTTTGCTCCCAAATTGAATG F<br>GGTTTTCCCAGTCACGACGATGCTTAGTGCAGCTTGTTC R | PET | 165–191 | $(TC)_{18}CG(TG)_{16}$ | AB689203 |

**Notes.**

F, forward; R, reverse.

## DNA extraction and amplification

Silica gel-dried leaves (5 cm in length) from each shoot were ground using a mortar and pestle. Genomic DNA was extracted using DNeasy plant mini kit (Qiagen®, Hilden, Germany) following the manufacturer's protocol. Eight microsatellite loci (Eaco_001, Eaco_009, Eaco_019, Eaco_050, Eaco_051, Eaco_052, Eaco_054, Eaco_055) developed by *Nakajima et al. (2012)* were used to score individual genotypes (Table 2). Forward primers were each labeled with a fluorescent dye 6-FAM, NED, PET or VIC (Applied Biosystems®, Foster City, CA, USA) (Table 2).

Polymerase Chain Reaction (PCR) was performed into two ways. (i) five loci (Eaco_001, Eaco_009, Eaco_019, Eaco_051, Eaco_054) were amplified using the Qiagen multiplex PCR Kit in a 10 µl total reaction containing 3 µl ddH$_2$O, 5 µl of 2× PCR Master Mix, 1 µl of 0.2 µM primer mix and 1 µl template DNA. PCR cycling was carried out for 5 min at 95 °C, followed by 35 cycles of 30 s at 95 °C, 1.5 min at 57 °C and 30 s at 72 °C with an extension of 30 min at 60 °C in the final cycle. (ii) three loci (Eaco_050, Eaco_052, Eaco_055) were amplified in a 20 µl total reaction containing 7.8 µlddH$_2$O, 2 µl of 1× PCR gold buffer, 2 µl MgCl$_2$ (2.5 mM), 2 µl dNTPs (200 µm), 1.5 µl of each 0.75 µm primer, 1U taq polymerase (Applied Biosystems®, Foster City, CA, USA) and 3 µl DNA template. PCR cycling was carried out for 15 min at 95 °C, followed by 32 cycles of 30 s at 94 °C, 1.5 min at 58 °C and 60 s at 72 °C with an extension of 30 min at 60 °C in the final cycle. All PCRs were performed on a 2720 Thermal Cycler (Applied Biosystems®, Foster City, CA, USA). GeneScan™ 500 LIZ® (Applied Biosystems®, Foster City, CA, USA) was used as an internal lane standard and PCR products were sent to UC Berkeley DNA Sequencing Facility, USA, where they were run on an ABI 3130 Xl automated DNA sequencer (Applied

Biosystems®, Foster City, CA, USA). Individual genotypes were scored using Geneious ver. 7.0.6 (*Kearse et al., 2012*).

## Data analysis

### *Genetic diversity and Hardy-Weinberg equilibrium (HWE)*

The number of alleles (A), observed heterozygosities ($H_O$) and expected heterozygosities ($H_E$) were calculated using Genalex ver. 6.5 (*Peakall & Smouse, 2012*). Departures from Hardy–Weinberg equilibrium (HWE) for each locus in all populations was computed via the Markov Chain method (dememorization = 1,000, batch = 100, iterations per batch = 1,000) using Genepop on the web (*Jennings & Blanchard, 2004*). Levels of statistical significance were corrected according to a false discovery rate (FDR) correction (*Benjamini & Hochberg, 1995*). Microchecker (*Van Oosterhout et al., 2004*) was used to check for existence of null alleles and genotypic scoring error due to stuttering with 1,000 randomizations and a 95% confidence level. Null allele frequencies were estimated for each locus and population by the expectation maximization (EM) algorithm (*Dempster, Laird & Rubin, 1977*) as implemented in FreeNA (*Chapuis & Estoup, 2007*).

### *Population structure*

Genetic structure among populations was assessed in multiple ways. First, genetic differentiation was estimated between pairs of populations with the estimator θ (*Weir & Cockerham, 1984*) as implemented in Arlequin ver. 3.5.1 (*Excoffier & Lischer, 2010*). Second, we inferred the phylogenetic relationship among populations using Poptree2 (*Takezaki, Nei & Tamura, 2010*) with the Neighbor Joining method (*Saitou & Nei, 1987*) using Nei's $D_A$ distance (*Nei, Tajima & Tateno, 1983*) and 1,000 bootstrapping replicates. Tree topology was rooted using Mega 5 (*Tamura et al., 2011*).

Third, we used Structure 2.3.4 (*Pritchard, Stephens & Donnelly, 2000*) to infer population clustering and assign individuals to groups based on their microsatellite genotype. Five replicate runs were conducted for each *K* between 1 and 10 using an admixture model and assuming correlated allele frequencies (*Falush, Stephens & Pritchard, 2003*). Each run consisted of a burn-in of 20,000 followed by 100,000 steps of Markov Chain Monte Carlo (MCMC) sampling. The best *K* was determined using the Δ*K* method (*Evanno, Regnaut & Goudet, 2005*) as implemented in Structure Harvester (*Earl & VonHoldt, 2012*). Run data were merged by Clumpp (*Jakobsson & Rosenberg, 2007*) and population structure then displayed graphically using Distruct (*Rosenberg, 2004*).

Finally, we used a Mantel test to evaluate the statistical significance of isolation-by-distance (IBD). To control for the potential existence of hierarchical population structure (*Meirmans, 2012*), we also estimated partial Mantel tests controlling for geographic distance and the clustering groups previously identified, which best explained the genetic structure. Pairwise genetic distances ($F_{ST}$) among localities were imported from Arlequin. A geographic distance matrix was generated using the PATH tool implemented in Google Earth (Google Earth Plus for Windows) that calculates the shortest distance by sea. In the third matrix used to run the partial Mantel test, populations belonging to the same clustering group were coded with 0, and populations belonging to different clustering

groups with 1. Both Mantel and partial mantel tests were performed using Isolation-by-distance Web Service (IBDWS) version 3.23 (*Jensen, Bohonak & Kelley, 2005*) with 10,000 randomizations.

## RESULTS

### Genetic diversity and Hardy-Weinberg equilibrium

Two loci (Eaco_052 and Eaco_050) were discarded after initial examination because of quality control issues. Eaco_052 had only a single allele in all populations (monomorphic), while Eaco_050 was not successfully amplified in most samples. Thus, only six loci (Eaco_001, Eaco_009, Eaco_019, Eaco_051, Eaco_054, and Eaco_055) were used for further analysis (Table 3). All samples were successfully amplified for all six loci, except a single sample from ANS for which Eaco_051 could not be amplified.

A total of 89 alleles were detected across these six microsatellite loci, ranging from one allele at the locus Eaco_019 in BM, KJ, BK, and ANS populations to 13 alleles at locus Eaco_054 in the BM population. The mean number of alleles per locus ranged from 1.57 to 9.57, and the mean number of alleles per population ranged from 4.33 to 7.00 (Table 2). The BM population had the highest average number of alleles, while the lowest was found in the KJ population. Observed heterozygosities ranged from 0.434 to 0.615 and expected ($H_E$) from 0.458 to 0.605, respectively. The TD population had the highest $H_E$ value (0.605), while the ANS showed the lowest $H_E$ (0.458).

Deviation from HWE is evidence that simple genetic models may not be appropriate (*Waples & Allendorf, 2015*). Hardy-Weinberg equilibrium tests revealed that five loci (Eaco_001 in TD and NK, Eaco_009 in KJ and ANS, Eaco_051 in KJ, Eaco_054 in TD, NK, ANS and Eaco_055 in ANS) deviated significantly from Hardy-Weinberg expectations ($P < 0.05$) prior to a multiple test correction with a false discovery rate. After correction for multiple tests, only two loci significantly deviated from Hardy-Weinberg equilibrium (Eaco_001 and Eaco_054).

Microchecker detected homozygote excess at locus Eaco_054 in TD and ANS and at locus Eaco_001 in TD and NK, likely a result of null alleles. Null alleles may also explain the significant deviation from Hardy-Weinberg expectations at these loci (Eaco_054 and Eaco_001), although inbreeding due to population substructure (Wahlund effect) and natural selection are also possible. Null alleles were potentially implicated in 16 out of 42 locus-population combinations, with an estimated frequency of <0.2 (Table 4). Null alleles may cause an overestimation of genetic differentiation (*Chapuis & Estoup, 2007*), but the effect is not considerable when their frequency is lower than 0.2 (*Carlsson, 2008*). To test the effects of null alleles to our datasets, we conducted pairwise $F_{ST}$ tests by including or excluding locus Eaco_054 (in which three out of seven locations were deviated from HWE after FDR correction); removing Eaco_054 did not change the results of pairwise comparisons, therefore all loci were included in further analysis. Individual genotypes for these loci are reported in Data S1.

**Table 3  Summary of genetic diversity for *E. acoroides*. Genetic diversity was inferred from the numbers of alleles (A), observed heterozygosities ($H_O$) and expected heterozygosities ($H_E$).** Numbers in bold indicate significant deviation from Hardy–Weinberg equilibrium prior to multiple test correction and bold number with ∗ (asterix) indicate significant deviation at $P < 0.05$ after corrections for false discovery rates (*Benjamini & Hochberg, 1995*).

| Loci | Locations | | | | | | | Total alleles | Mean A/locus |
| --- | --- | --- | --- | --- | --- | --- | --- | --- | --- |
| | TD | PR | NK | BM | KJ | BK | ANS | | |
| **Eaco_001** | | | | | | | | | |
| A | 3 | 2 | 8 | 5 | 2 | 3 | 2 | 10 | 3.57 |
| $H_O$ | 0.185 | 0.389 | 0.400 | 0.467 | 0.417 | 0.429 | 0.355 | | |
| $H_E$ | 0.427 | 0.375 | 0.611 | 0.417 | 0.375 | 0.489 | 0.331 | | |
| P | **0.002*** | 1.000 | **0.000*** | 0.834 | 1.000 | 0.274 | 1.000 | | |
| **Eaco_009** | | | | | | | | | |
| A | 5 | 5 | 2 | 6 | 6 | 6 | 6 | 10 | 5.14 |
| $H_O$ | 0.667 | 0.667 | 0.033 | 0.600 | 0.417 | 0.786 | 0.613 | | |
| $H_E$ | 0.658 | 0.650 | 0.095 | 0.660 | 0.549 | 0.677 | 0.592 | | |
| P | 0.940 | 0.952 | 0.049 | 0.331 | **0.017** | 0.674 | **0.026** | | |
| **Eaco_019** | | | | | | | | | |
| A | 2 | 2 | 3 | 1 | 1 | 1 | 1 | 3 | 1.57 |
| $H_O$ | 0.111 | 0.500 | 0.467 | 0.000 | 0.000 | 0.000 | 0.000 | | |
| $H_E$ | 0.105 | 0.461 | 0.376 | 0.000 | 0.000 | 0.000 | 0.000 | | |
| P | 1.000 | 1.000 | 0.610 | N.A | N.A | N.A | N.A | | |
| **Eaco_051** | | | | | | | | | |
| A | 9 | 4 | 9 | 9 | 7 | 10 | 8 | 19 | 8.00 |
| $H_O$ | 0.852 | 0.722 | 0.933 | 0.667 | 0.917 | 0.905 | 0.633 | | |
| $H_E$ | 0.811 | 0.619 | 0.808 | 0.757 | 0.759 | 0.826 | 0.724 | | |
| P | 0.554 | 0.861 | 0.504 | **0.022** | **0.009** | 0.905 | 0.189 | | |
| **Eaco_054** | | | | | | | | | |
| A | 10 | 8 | 10 | 13 | 6 | 12 | 8 | 29 | 9.57 |
| $H_O$ | 0.630 | 0.667 | 0.800 | 0.833 | 0.792 | 0.810 | 0.290 | | |
| $H_E$ | 0.858 | 0.789 | 0.839 | 0.792 | 0.658 | 0.841 | 0.388 | | |
| P | **0.000*** | 0.171 | **0.003*** | 0.246 | 0.725 | 0.880 | **0.006*** | | |
| **Eaco_055** | | | | | | | | | |
| A | 6 | 6 | 9 | 8 | 4 | 7 | 5 | 18 | 6.43 |
| $H_O$ | 0.704 | 0.667 | 0.700 | 0.800 | 0.667 | 0.762 | 0.710 | | |
| $H_E$ | 0.771 | 0.657 | 0.823 | 0.699 | 0.656 | 0.747 | 0.710 | | |
| P | 0.130 | 0.427 | 0.075 | 0.889 | 0.143 | 0.090 | **0.022** | | |
| **All pop.** | | | | | | | | | |
| A | 5.83 | 4.50 | 6.83 | 7.00 | 4.33 | 6.50 | 5.00 | 14.83 | 5.71 |
| $H_O$ | 0.525 | 0.602 | 0.556 | 0.561 | 0.535 | 0.615 | 0.434 | | 0.547 |
| $H_E$ | 0.605 | 0.592 | 0.592 | 0.554 | 0.499 | 0.597 | 0.458 | | 0.557 |

**Table 4** Frequency of null alleles per locus per population for *E. acoroides*.

| Loci | Population | | | | | | |
| --- | --- | --- | --- | --- | --- | --- | --- |
| | TD | PR | NK | BM | KJ | BK | ANS |
| Eaco_009 | 0.000 | 0.000 | 0.110 | 0.043 | 0.077 | 0.000 | 0.008 |
| Eaco_019 | 0.000 | 0.000 | 0.000 | 0.001 | 0.001 | 0.001 | 0.001 |
| Eaco_054 | 0.121 | 0.041 | 0.000 | 0.000 | 0.000 | 0.011 | 0.068 |
| Eaco_001 | 0.174 | 0.000 | 0.145 | 0.000 | 0.000 | 0.038 | 0.000 |
| Eaco_051 | 0.000 | 0.000 | 0.000 | 0.029 | 0.000 | 0.000 | 0.048 |
| Eaco_055 | 0.070 | 0.000 | 0.067 | 0.000 | 0.000 | 0.002 | 0.056 |

**Table 5** Pairwise $F_{ST}$ values (below diagonal) and geographic distances (above diagonal).

| | TD | PR | NK | BM | KJ | BK | ANS |
| --- | --- | --- | --- | --- | --- | --- | --- |
| TD | – | 37.3 | 1,418 | 850 | 456 | 328 | 1,000 |
| PR | 0.127[*] | – | 1,456 | 834 | 419 | 317 | 986 |
| NK | 0.290[*] | 0.302[*] | – | 1,706 | 1,875 | 1,716 | 2,050 |
| BM | 0.235[*] | 0.270[*] | 0.301[*] | – | 1,000 | 511 | 356 |
| KJ | 0.225[*] | 0.273[*] | 0.338[*] | 0.247[*] | – | 525 | 1,095 |
| BK | 0.175[*] | 0.203[*] | 0.214[*] | 0.127[*] | 0.247[*] | – | 682 |
| ANS | 0.298[*] | 0.348[*] | 0.359[*] | 0.293[*] | 0.239[*] | 0.303[*] | – |

**Notes.**
[*]$P < 0.001$.

## Population structure

Pairwise $F_{ST}$ values ranged from 0.127 to 0.359. $F_{ST}$ were significant between all pairs of samples with $P < 0.001$ (Table 5). The largest genetic differentiation was found between samples from NK and ANS ($F_{ST} = 0.359$), while the smallest pairwise difference was found between TD and PR and between BM and BK ($F_{ST} = 0.127$).

The Neighbor Joining method based on $D_A$ distance identified three major clusters (Fig. 2). Cluster 1 (NK) was genetically distinct from all other populations. Cluster 2 consists of two populations in Java (TD and PR) and two populations in Sumatra (BK and BM). The highest bootstrap support was found between TD and PR. In contrast, two populations, KJ and ANS showed discordance between genetic and geographic distance (cluster 3).

The $\Delta K$ test in Structure (*Pritchard, Stephens & Donnelly, 2000*) indicated the maximum value of $\Delta K$ at $K = 3$ with a secondary peak at $K = 6$ and $K = 7$ (Fig. S1). At $K = 3$, the Structure analysis mirrors the groupings seen in Neighbor Joining tree, with the same three major clusters (NK; KJ, ANS; and TD, PR, BM, BK) (Fig. 2). Isolation-by-distance (IBD) revealed significant correlation between genetic differentiation and geographic distance across all pairs of samples, with $P = 0.003$ (Fig. S2). Due to the existence of hierarchical structuring, a partial Mantel test was conducted in which population NK was excluded, given the scale of its genetic and geographic distance from all other sites. A partial Mantel

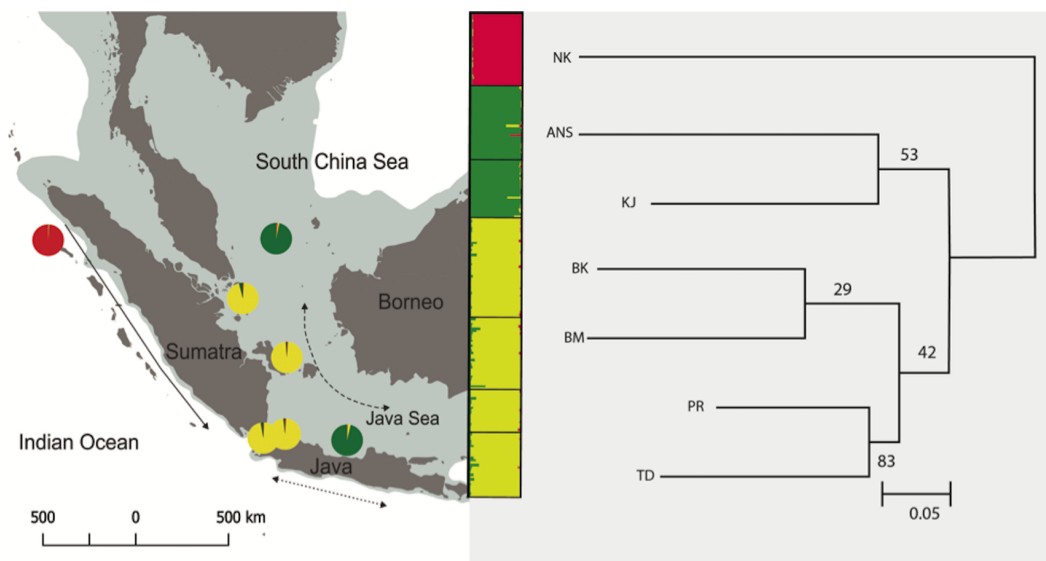

**Figure 2** **Neighbor Joining (NJ) tree based on $D_A$ distance is consistent with Structure (clustering inset) showing that all populations were divided into three clusters (NK/TD, PR, BK and BM/ANS and KJ).** Each color represents one cluster in the Structure analysis and the length of each colored segment shows the proportion of coanscestry as a measure of membership in that cluster. Bootstrap values on the NJ tree are shown besides the node as the number replicates, out of 100, in which the cluster was formed.

performed on the remaining Java and Natuna Sea locations indicated that IBD is present across these sites ($p = 0.03$, Fig. 3)

## DISCUSSION

The present study is the first report of genetic diversity (measured as $H_O$, $H_E$, and the number of alleles), genetic differentiation and phylogeographic patterns of *E. acoroides* in the Indo-Malay Archipelago. Despite the limited number of loci (six microsatellites) used in this study, high observed heterozygosity ($H_O$) is found in all populations. Our results are consistent with the general patterns of limited dispersal and restricted gene flow at various geographical scales suggested by previous studies (*Lacap et al., 2002*; *Nakajima et al., 2014*). However, our results add to previous studies by not only confirming population differentiation between the Indian and Pacific Ocean but also, reveals surprising and previously unknown structure across the Java and the Natuna Sea.

### Genetic diversity

Genetic diversity is closely related to survivorship, resistance, and resilience of any individual or population to disturbance (*Williams, 2001*; *Hughes & Stachowicz, 2004*). Low genetic diversity can reduce the fitness of individuals, and lead to population extinction (*Spielman, Brook & Frankham, 2004*). In recent years, genetic diversity plays a central role in predicting the ability of individuals or populations to survive environmental change (*Hughes & Stachowicz, 2004*), and global climate change (*Ehlers, Worm & Reusch, 2008*).

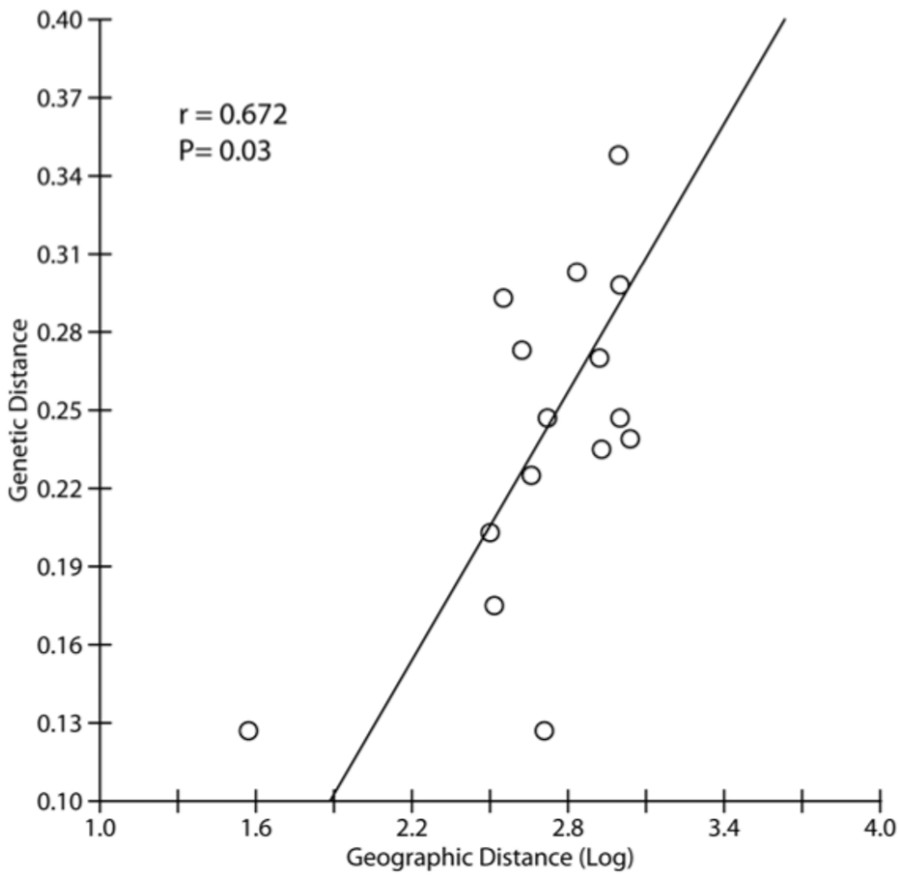

**Figure 3** Correlation between genetic and geographic distance used partial Mantel test based on $F_{ST}$ and genetic distance excluding NK, the most divergent population in all analyses.

Our study revealed that the highest genetic diversity was observed in the central part of Indonesia (BK and PR), while the northernmost population (ANS) possessed lower observed heterozygosity. Sample collection at BK and PR was conducted in undisturbed area where the seagrasses growth in a dense and continues bed (ING Putra & HH Madduppa, pers. comm., 2014). Seagrasses in undisturbed areas tend to be more genetically diverse (*Larkin et al., 2006*). Although ANS showed the lowest observed heterozygosity, this may not related to their local environment because sampling location was also conducted in undisturbed area. Microchecker results indicated that ANS had an excess of homozygosity, a potential indicator of inbreeding. Inbreeding mostly occurs in clonal organisms and reduces diversity by reducing the effective frequency of recombination throughout the genome (*Charlesworth, 2003*).

Previous studies of *E. acoroides* in Lembongan (Bali) and Waigeo (Papua), found that observed heterozygosities were 0.436–0.582, respectively (*Pharmawati et al., 2015*). Likewise, using nine microsatellite loci, *Nakajima et al. (2014)* found observed heterozygosities of *E. acoroides* ranging from 0.165–0.575 at three locations (Japan, China, and Philippines). Observed heterozygosities of *E. acoroides* varied from 0.100 to 0.567 in

China (*Gao et al., 2012*), highlighting that our observed values are quite similar to those reported from other sites. The differences in reported heterozygosity ranges between studies could arise due to various factors such as natural selection, variation in the mutational rate, and the number and nature of the loci used.

Seagrasses in the Indo-Malay Archipelago, as in other parts in the world, remain subject to overexploitation with conservation efforts largely marginalized. Although the potential decrease of genetic diversity in seagrasses is quite high due to meadow fragmentation (*Vermaat et al., 2004*) and extensive clonal growth (*Procaccini, Olsen & Reusch, 2007*), *E. acoroides* appears to retain considerable genetic variation through sexual reproduction. Sexual reproduction contributes significantly to maintaining genetic variation in clonal organisms (*Rollon, Cayabyab & Fortes, 2001*). The available life history information also suggests that most seagrass species are predominantly outcrossing (*Reusch, 2001*).

## Population genetic structure

Pairwise $F_{ST}$ showed significant genetic differentiation between all pairs of populations. Moreover, Isolation-by-distance (IBD) showed a significant correlation between genetic differentiation and geographic distance, even when the most distant and differentiated population (NK) is excluded. These results indicate that there is low gene flow between populations, possibly due to limited dispersal potential of *E. acoroides*. For example, *Lacap et al. (2002)* showed that pollen dispersal of *E. acoroides* is limited over a small spatial scale, usually less than 10 km. The low gene flow discovered in the present study and from previous studies (e.g., *Nakajima et al., 2014*) are consistent with direct observations made on fruit and seed floating time, where median flotation periods were 7 days for fruit and 0.5 h for seeds, during which time they can cover distances of 41 km and 0.1 km, respectively (*Lacap et al., 2002*). After buoyancy is lost for seeds, further seed dispersal probably continues for 2 to 5 days, but the maximum distance was only 204 cm (*Lacap et al., 2002*).

Limited dispersal distances are also indicated in other seagrass species such as *Thalassia hemprichii* (*Lacap et al., 2002*) and *Thalassia testudinum* (*Davis, Childers & Kuhn, 1999*). Likewise, *Hosokawa et al. (2015)* found that seeds of *Zostera marina* do not disperse far from the parent bed. Thus, such local genetic differentiation may be common among seagrass species, although *Zostera noltii* shows a large geographic area of apparent genetic homogeneity (*Coyer et al., 2004*).

Phylogeographic relationships among *E. acoroides* suggest its partitioning into three major groups, consistent with Bayesian clustering analysis (Fig. 2). Clustering output indicated secondary peaks at $K = 6$ and $K = 7$. These results indicate the presence of further substructure similar to that found in $F_{ST}$ analysis. At $K = 6$, all populations are separated except for populations in TD and PR, while at $K = 7$, all populations seem to be genetically distinct. NK, the most geographically distant population, showed distinct genetic structure from the six other sites, and represents the western Indonesian population (Indian Ocean population). Differentiation of the western Indonesian (Indian population) is common in other organisms and is believed to be largely a result of Pleistocene vicariance (*Barber et al., 2000*; *Crandall et al., 2008*; *Yasuda et al., 2009*). Low sea levels during the Pleistocene

would result in the emergence of the Sunda Shelf, which presumably isolates *E. acoroides* by preventing dispersal. Studies on marine plants such as mangroves, *Bruguiera gymnorrhiza* (*Urashi et al., 2013*), genus *Ceriops* (*Huang et al., 2008*), and the seagrass *H. ovalis* (*Nguyen et al., 2014*) all showed the Sunda Shelf as being a geographical barrier between the Indian and Pacific Ocean populations. Further, similar genetic patterns consistent with Pleistocene isolation have also been reported for marine animals such as in false clownfish *Amphiprion ocellaris* (*Nelson et al., 2000*), mantis shrimp *Haptosquilla pulchella* (*Barber et al., 2002*), the coral reef fish *Caesio cuning* (*Ackiss et al., 2013*), the giant clam *Tridacna crocea* (*DeBoer et al., 2014*), and five commercially important pelagic fishes (*Auxis thazard, Euthynnus affinis, Katsuwonus pelamis, Rastrelliger kanagurta,* and *Scomberomorus commerson*) (*Jackson et al., 2014*) respectively.

The geographical partitioning of genetic groups observed in the present study may be a result of various events in the past, but the effect of recent physical oceanography cannot be ignored (*DeBoer et al., 2014*). The importance of currents in shaping the genetic structure have been reported on both large and small geographical scales (*Barber et al., 2002*; *Galarza et al., 2009*; *Yasuda et al., 2009*; *Zhan et al., 2009*; *Wee et al., 2014*). In this region, the South Java Current (SJC) is a unidirectional current, flowing down the west coast of Sumatra (Fig. 1). This current probably acts as a barrier to the dispersal of *E. acoroides* between NK and the other sites.

The genetic structuring of cluster 2 in this study mirrors the genetic structure found in the genus *Hippocampus* (*Lourie, Green & Vincent, 2005*) and *Nerita albicilla* (*Crandall et al., 2008*) respectively. However, no phylogeographic break was found across Java Sea, which is in contrast to previous studies carried out in stomatopods (*Barber et al., 2002*; *Barber, Erdmann & Palumbi, 2006*). It is possible that *E. acoroides* found appropriate habitat across the Seribu, Bangka, and Batam Islands and used these islands as stepping stones for long distance dispersal. Alternatively, chance may play a role such that each species responds to history in a slight different manner, resulting in a variety of genetic patterns (*Bowen et al., 2016*). If this were the case, we would expect a shared pattern of genetic structure to emerge if enough species were sampled, as has been reported recently in Hawaii (*Toonen et al., 2011*; *Selkoe et al., 2014*; *Selkoe et al., 2016*).

The alternating monsoon in Indonesia also likely plays an important role in driving dispersal in the region, and is a possible confounding factor for estimating gene flow. Alternating monsoons generate two Seasonally Reversing Currents (SRC) across the Java, Natuna, and South China Seas (Fig. 1). These large storms and reversing currents could also create a complexity of local oceanographic features. Although, previous research did not find evidence of any relationship between genetic patterns and seasonal currents in the Sunda Shelf region for clownfish (*Nelson et al., 2000*), our results presented here indicates that SRC may play an important role in the dispersal of *E. acoroides*.

Finally, although we found significant isolation-by-distance overall, two sites (KJ and ANS) showed high genetic similarity despite being geographically distant from one another. This genetic pattern may also have been influenced by sea level changes during the Pleistocene. When the sea level was at 30 m Below Present Level (BPL), land bridges connecting Sumatra and Borneo via Bangka Belitung seem to have been lost (*Voris, 2000*).

Thus, both the Java and Natuna Seas became connected and this connection may have facilitated gene flow between ANS and KJ. At more recent timescales, oceanographic processes such as SRC may also create and maintain genetic structure between KJ and ANS. Seagrass populations in the Natuna Sea could be originating from a different source. *Kool et al. (2011)* suggest that areas in the Natuna Sea are supplied by larval transport from the South China Sea. Other studies showed that populations in the Natuna Sea may also receive propagules from the Andaman Sea via the Malaka Strait following the ice retreat after LGM (*Timm & Kochzius, 2008*; *Wee et al., 2014*). Due to observed differentiation between populations in BM and ANS, we suspect these populations might originate from different areas. The existence of biophysical barriers separating western and eastern part of the Natuna Sea might also responsible for the population differentiation in this area (*Treml et al., 2015*). However, in order to better understand the origins of the populations in BM and ANS, additional sampling sites from the Malaka Strait and the South China Sea are necessary.

## Implications for restoration and management

Seagrass beds provide important habitat for a wide variety of organisms in marine ecosystems (*Hemminga & Duarte, 2000*). However, they are declining globally due to human activity and climate change (*Duarte, 2002*; *Waycott et al., 2009*). Restoration of seagrass beds is underway in many sites worldwide (*Van Katwijk et al., 1998*; *Park & Lee, 2010*). Recently, it has been found that genetic diversity plays an important role for seagrass resilience and restoration (*Ehlers, Worm & Reusch, 2008*; *Reynolds, McGlathery & Waycott, 2012*).

Here we show there is limited connectivity and significant genetic differentiation between all sampling sites. This study also revealed there are three different cluster based on genetic distance and Bayesian clustering. Therefore, we recommend resource managers treat populations across our sampling region as three Management Units (MUs) to maintain the unique genetic characteristics of each region to support seagrass conservation goals. Any restoration activities conducted using local stocks (the same structure cluster) would be beneficial because their genetic diversities have been assessed. Population with higher genetic diversity increases the fitness and reproductive success of the transplants (*Williams, 2001*). Genetic diversity also contributed to the resistance of seagrass beds to various disturbances (*Hughes & Stachowicz, 2004*); thus, high diversity creates more resilient ecosystems, and provides more ecosystem services (*Reynolds, McGlathery & Waycott, 2012*).

Introducing new genotypes through the transplanting seed or plants to other sites should be avoided. Introduction of new alleles that are distinct from native populations would lead to lower survivorship if not locally adapted (*Williams, 2001*). When migrants are translocated or transferred from one population to another, they may mate with native population and produce offspring that are less fit (*outbreeding depression*). Although, this has not been documented or recorded in seagrasses so far, this event has been found in wild populations of scarlet gilia (*Ipomopsis aggregata*) (*Waser, Price & Shaw, 2000*).

## CONCLUSIONS

Despite the loss of seagrass habitat, these plants manage to maintain good genetic diversity and heterozygosity. Genetic diversity analysis revealed high levels of heterozygosity in all sampling sites. Our results also indicate low gene flow among sites of *E. acoroides,* as evidenced by significant genetic differentiation among sites, and significant isolation-by-distance. Strong genetic differences were observed between the Indian Ocean population (NK) and all six others. These differences probably represent Pleistocene vicariance between Indian and Pacific populations, consistent with similar patterns in other marine flora and fauna. Seasonally variable currents and extreme storm events may play a role in long-distance dispersal of *E. acoroides* because the phylogeographic pattern connecting the two peripheral locations that are outliers from the IBD pattern match these alternating currents instead. Our data will help to guide management and restoration efforts by confirming high heterozygosity among most local populations, and highlighting boundaries between which translocations are most desirable. In conclusion, when planning a restoration project, genetically guided criteria should be used in the selection of donor material. Therefore, management plans for seagrass in Indonesia should add such genetic evaluation of seagrasses to conservation and restoration planning.

## ACKNOWLEDGEMENTS

We thank Dondy Arafat (Marine Biodiversity and Biosystematics Laboratory, IPB) for collecting samples in Batam, Ibu Widiastuti (Diponegoro University) for collecting samples in Karimun Jawa, Khalidin for sample collection in Aceh, Dedi, Gugun and Okto for their assistance with sample collection in Bangka. We would like to also express our gratitude to Indonesian Biodiversity Research Center (IBRC) who had facilitated all the research works, provided research networks, organized sampling collection and provided laboratory tools and equipments. This specifically goes to Aji Wahyu Anggoro, Dita Cahyani and Prof. IGN Mahardika for their contribution in assuring the research running well. We would like to also thank Samsul Bahri, Astria Yusmalinda, Rizki Wulandari, Andrianus Sembiring, Masriana, Eka Maya Kurniasih, Dian Pertiwi and Angka Mahardini from IBRC for their help in both the laboratory and field work. We would also like to extend our gratitude to editor and reviewers, and Joana Dias (Curtin University, Australia) for comments, reviews and suggestions on the manuscript.

### Funding

This study is mainly funded by PEER (the Partnerships for Enhanced Engagement in Research, No: PGA-2000003438) funded by United States Agency for International Development (USAID) and the National Science Foundation (NSF) in partnership with NSF-PIRE Program. This was also funded by the government of Indonesia through the Indonesia Endowment Fund for Education (LPDP). There was no additional external

funding received for this study. The funders had no role in study design, data collection and analysis, decision to publish, or preparation of the manuscript.

### Grant Disclosures

The following grant information was disclosed by the authors:

PEER (the Partnerships for Enhanced Engagement in Research): PGA-2000003438.

United States Agency for International Development (USAID).

National Science Foundation (NSF).

Indonesia Endowment Fund for Education (LPDP).

### Competing Interests

The authors declare there are no competing interests.

### Author Contributions

- I Nyoman Giri Putra and Hawis Madduppa conceived and designed the experiments, performed the experiments, analyzed the data, contributed reagents/materials/analysis tools, prepared figures and/or tables, authored or reviewed drafts of the paper.
- Yuliana Fitri Syamsuni analyzed the data, contributed reagents/materials/analysis tools, authored or reviewed drafts of the paper.
- Beginer Subhan performed the experiments, analyzed the data, authored or reviewed drafts of the paper.
- Made Pharmawati conceived and designed the experiments, analyzed the data, contributed reagents/materials/analysis tools, authored or reviewed drafts of the paper.

### Data Availability

The raw data has been supplied as Supplemental Dataset.

### Supplemental Information

Supplemental information for this article can be found online at http://dx.doi.org/10.7717/peerj.4315#supplemental-information.

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
