# Peer review of "Strong genetic differentiation in tropical seagrass Enhalus acoroides (Hydrocharitaceae) at the Indo-Malay Archipelago revealed by microsatellite DNA"

_PeerJ, doi:10.7717/peerj.4315_

## Round 0.1 · original submission · Major Revisions

I now have two reviews back that are quite consistent in the comments and suggestions for improvement of the manuscript prior to it being acceptable for publication. There are 5 main issues that the referees bring up with the submission, and will need to be addressed prior to sending the paper back to the referees for reassessment. The first is the stylistic and English grammar issues. Both comment on this issue and it is a requirement of PeerJ for a manuscript being acceptable for publication. The major second issue that both bring up is the IBD analyses and Meirman's k-means clustering approach to determine whether it is truly IBD or not. The third issue is the family-wise error rate and use of False Discovery Rate (FDR) correction rather than Bonferonni. This is a good point, and an effort to set the alpha to 5% overall is warranted. The fourth major point raised by referees has to do with the quality control of the loci, nulls, linkage and possible selection issues. The simplest way I see to deal with this issue is to include both a locus-by-locus analysis and jackknife across loci to show the impact of individual loci on your overall pattern and the strength of the conclusions. Finally, I agree with the reviewers that the AMOVA is redundant if you simply test the groups you find in the other analyses. In my opinion, the better use of AMOVA is to test the a-priori hypotheses that you had going into the study rather than the post-hoc testing of the groups you find in the study. I would suggest that you either set it up as the suite of biological or conservation hypotheses you wish to test and then use AMOVA to evaluate support for each, or delete the currently redundant analysis.

None of these criticisms seem overly difficult to address, and I expect that you should be able to revise and improve the manuscript based on the useful feedback of these referees. I look forward to seeing your revised manuscript.

Reviewer 1 ·

Basic reporting

The manuscript is riddles with inconsistencies in both grammar and spelling. Once the appropriate revisions have been made, the MS will need to go to a native English speaker for rewriting and checking. More detail is given in the 'general comments to authors section'.

Experimental design

ok

Validity of the findings

some misgivings in places - see general comments.

Additional comments

Review of “Strong genetic differentiation in tropical seagrass Enhalus acoroides (Hydrocharitaceae) at the Indo-Malay Archipelago revealed by microsatellite DNA” by Putra et al.
This paper examines the genetic diversity and phylogeography of the seagrass, Enhalus acoroides, in the Indo-Malay Archipelago. The major conclusions of this paper are that this seagrass exhibits a high level of genetic diversity and structure. Three genetic clusters of populations were identified, with significant differentiation between populations on either side of the Sunda shelf. Several explanations are given for the pattern of genetic structure observed. Overall, I feel that this paper makes a meaningful contribution to uncovering phylogeographic patterns and barriers in the region, especially as phylogeographic studies on these important foundational species are absent from this region. I have no major comments on the science, but there are many grammatical errors which detract from the overall impression of the paper. I suggest that this paper is publishable, as long as the authors conduct an extensive grammatical revision and address the comments below. The paper will need to be seen again by the reviewers before it can be accepted.
Major comments-
• An extensive grammatical revision will be required, specifically in the introduction and discussion, but also elsewhere. I have pointed out some of these as an example, but have not continued with grammatical editing past line 105.

• There is a lot of discussion around genetic diversity, heterozygosity in the study species and other species in the study region, and resilience to disturbance. However, you don’t actually draw any conclusions or suggest what the level of genetic diversity your species exhibits might mean. This would be useful here in order to put your discussion on ‘Implications for restoration and management’ later on into context.

• You will need to consider the role of animals on the transport of seeds or vegetative material, as this has been found in other seagrasses.

• Discussion on cluster 3 is lacking. Why do the populations in cluster 2 group separately from those in cluster 3? I understand that your major talking point is the high genetic diversity and genetic differentiation between the Indian and Pacific Oceans, but the differentiation between cluster 2 and 3 is an interesting finding that is not discussed and should be.

• Why did you only sample one population from the Indian Ocean? A more balanced sampling scheme would have given a better view of the overall picture.

• The flow of the ‘Implications for restoration and management’ section is very poor and should be improved.

• Following on from this – I do not think it is correct to be comparing genetic diversity estimates of plants and animals. Rather focus the discussion around other seagrass species – there are nunerous studies. Perhaps there are differences between tropical and temperate seagrass species that can be explored. This section requires rethinking. What does genetic diversity mean in the context of your sudy and why is it important? Detail this clearly.


• Isolation by distance – you report a strong signal of IBD, yet this might be as a result of you carrying ot your IBD analyses across regions with known genetic breaks. Meirmans (2012) wrote a great paper “The trouble with isolation by distance” that details why a break in gene flow can inflate a signal of IBD. You should test for IBD within each of your clusters separately, rather than across clusters if that is possible.
Minor comments- mainly grammatical
Line 19: ‘regarded as a barrier that isolates organisms of the Indian and…’
Line 22: ‘organisms ‘ plural
Line 33: ‘in shaping the genetic patterns of…’
Line 49-52: this sentence is a bit confusing and could be split up and clarified.
Line 53: ‘large land masses...’
Line 60 and 61: ‘populations’ plural, as you are talking about populations of different species
Line 61: Central rather than “middle”?
Line 63: ’a concordance in the geographical barrier…’
Line 67: ‘the Halmahera Eddy…’
Line 67: ‘are likely candidates shaping the biogeographic barrier…’
Line 72: Geological instead of ‘geographical’?
Line 74: divergence instead of ‘diversification’?
Line 78 ‘bivalves’ plural
Line 79: by ‘concordance’ do you mean they show the same pattern in terms of structure or genetic diversity, or do you actually mean that these populations are related/connected via gene flow. Please explain.
Line 83: ‘nuclear DNA markers…’
Line 84: missing full-stop after reference
Line 85: ‘using seagrasses is lacking.’
Line 89: ‘occurs’
Line 89: Include the common name for the species if you can.
Line 95: ‘fibrous strands (the remnants of old leaves).’
Line 96: ‘are capable of floating for up to 10.2 days…’
Line 98: ‘respectively (Lacap et al., 2002). This might limit…’
Line 99: ‘although, long distance…’
Line 100: ‘dispersal capacity…’
Line 102: ‘to find significant…’
Line 105: ‘used broadly in phylogeographic…’
Line 179: ‘NJ’ is this Neighbour joining? Either write out in full of define this abbreviation when you first introduce it.
Line 196-204: Maybe state why you are testing for deviation from HWE (you are testing whether your microsatellites are selectively neutral). Even though most analyses assume HWE, we know that HWE is not a realistic expectation (as you say, it can be influenced by population size etc.)
Line 216-221: Some repetition that could be removed.
Line 233: make the CT subscript.
Line 243: be more specific here. Did these studies look at a different region or different species?
Line 243-244: ‘The results presented here do not also reveal the…’ did you mean ‘do not only’?
Line 276: ‘all pairs of populations.’
Line 293: ‘NK, where the most geographically distant, showed distinct…’ rephrase this sentence, as it currently makes very little sense.
Line 295-297: you could say ‘Differentiation of the western Indonesian (Indian population) seems to be largely a result of…’ This acknowledges that there are other factors at play, such as current, which you discuss later.
Line 310: forcing of genetic homogeneity? I understand what you are getting at but this needs further explanation
Line 317: Try not to use ‘probably’ in this context. Rather say something like ‘it is possible that…’, ‘it is likely that…’ or ‘the seagrass may have…’
Line 320: how about ‘the alternating monsoon in Indonesia is also a possible confounding factor.’
Line 333: this contradicts what you said earlier about currents acting as dispersal barriers and the low dispersal capacity of this species
Line 337: ‘ and a high level of genetic differentiation…’
Line 340: ‘local stocks’ are you referring to the entire Indo-Malay Archipelago or a specific site?
Line 345: Does seagrass with a higher level of genetic diversity provide ‘more’ ecosystems services or ‘better quality’ ecosystem services?
Line 349: ‘mate’ is not a good choice of words to describe plant sexual reproduction

Reviewer 2 ·

Basic reporting

To the best of my knowledge, this manuscript adheres to policies and templates. In my opinion, this topic is a valuable contribution to PeerJ and the authors do a sufficient job at explaining the relevancy of this research.

The manuscript contains extensive grammatical mistakes that must be fixed. I am assuming these arise from the fact that the authors are writing English as a second language. The good news is that these mistakes are primarily tense-based (noun or verbs that should switch from plural-singular or vice versa) and minor spelling errors. I found very few sentences that were unclear or ambiguous in meaning despite these errors. I recommend the authors run this manuscript through a word processor with an English spell/grammar check such as Microsoft Word and work with PeerJ or a native English-speaking colleague to review the manuscript explicitly to correct these mistakes. In this review, I limit my suggested changes to the abstract as an example:

Line 19 – “ is regarded as [a] barrier that isolates organism[s] of [the] Indian and”
Line 20 – “biota from [this region] have found”
Line 21 – “, seemingly dependent on [] taxon and [] methodology. The [g]enetic “
Line 22 – “marine organism[s] in [this region] are related to several”
Line 23 – “ Shelf[] and recent physical”
Line 28 – “Both a [neighbor joining] tree based on”
Line 29 – “ E. acoroides.[] Further, AMOVA revealing significant partition[ing] of”
Line 32 – “such as [the] South Java Current and [the] Seasonal[y] Reversing”
Line 33 – “in shaping genetic pattern[s] [in] E. acoroides.”

Other basic reporting on the body of the manuscript:

Line 49-52 “Although several phylogeographic studies…from both oceans.” This sentence is unclear and needs to be reworded. Are the authors trying to say that only a few studies have tried to identify specific elements in the region that promote genetic structuring? The term “overlapping region” is not good here.

Line 79 – phylogeographic DIScordance?

Line 107 – these markers are co-inherited (not co-dominant, since they are not expressed)

Line 184-187 – this 3-sentence paragraph begins and ends with sentences ending with “used for further analysis” I recommend the authors change the first to “were discarded after initial examination” or something similar so that the sentences are not so repetitive.

Line 200-201 – “However, according to Micro-checker…null alleles” This sentence is unclear and needs to be reworded for clarification. As a side note which I will bring up again, this sentence is the only reported result from running Micro-checker and I recommend more information is provided.

Line 326-327 – inconsistent correlation between genetic and geographic distance for KJ and ANS? Where is this information? There are no details about this in either the IBD figure or the results section.

Basic reporting on figures & tables:

Figure 1 – I recommend putting an “Indonesia” label on the small inset and perhaps placing a simple box around the zoomed area so that readers not familiar with this area can orient themselves.

Figure 2 – Negative geographic distance is impossible. The x-axis should start at 0. Along with other comments that will follow, I think that Nakuri comparisons should be removed from this figure and that comparisons between sites of different clusters (2 & 3) should be filled (e.g. gray circles instead of open ones) so that we can see how this correlation is affected by genetic clustering.

Table 1 – This table lists the number of individuals collected at each site but the authors should also include how many of these individuals were successfully genotyped at the 6 loci used (unless they had success for all samples, in which case they should mention that somewhere in the manuscript).

Table 3 – As is, there are only site-by-site data. Are the alleles found at each site unique? Or are alleles of different loci shared among sites. This table should also include the total number of alleles at each locus. I also recommend putting all pre-corrected significant deviations from HWE in bold and using an * (asterix) to denote with ones were still significant after correction.

Table 4 – Nowhere in this manuscript are the actual geographic distances between sites reported. The upper-diagonal *** is a little unnecessary since all sites were significantly different from one another. This can simply be reiterated in the table legend. I suggest replacing the upper-diagonal with geographic distance.

Experimental design

To the best of my knowledge, this work is well within the scope of research published by PeerJ. The authors do a good job of defining their research questions, and their study species and sampling design is appropriate and relevant.

Methods:

A kit was used for the first multiplexed reaction of 5 primer sets, but individual reagents were used in the second reaction of 3 primer sets. I recommend reporting concentrations along with volumes for primers, dNTPs or other reagents not tagged by supplier (i.e. concentration not need for the Amplitaq Gold/buffer from Applied Biosystems) for this second set so that these methods can be replicated.

Line 138-139 “GeneScan 500 LIZ…line standard.” I recommend moving this sentence in front of the one previous to keep the order chronological since size standards are added before sequencing occurs.

Data Analysis:

The greatest issue I have with the manuscript as it stands now is whether or not the 6 loci successfully amplified are neutral markers or influenced by some sort of selection. I am not confident that the authors have fully explored the data from these loci to properly determine whether these are markers that are neutral (and therefore useful in assumptions regarding passive seed transport and gene flow) or whether these markers are linked to sites that may be under selection and are therefore under the influence of local adaptation.

The red flag which makes me very closely examine the analyses of the authors is that when they ran pairwise Fsts, every site is significantly different from every other site. This could be an entirely true phenomenon based on highly restricted dispersal. But, this could also be a false signature driven by one or two of their 6 loci that might be influenced by selection or other issues that arise with a problematic locus (allelic dropout, etc). To address this issue, I suggest the authors start with the following:

1. The authors applied a very conservative method of correcting their p-values (Holm Bonferroni) for HWE tests which left anything with a p-value of < 0.000 significant (2 tests). When I played around with using the Benjamini-Hochberg (1995) method of correcting FDR on the author’s p-values (a method shown to correct for type I error while creating fewer type II errors), any p-value < 0.0065 remained significant (5 tests). Additionally, using the Holm Bonferroni method at each locus (m=7 instead of m=38/42) the same 5 tests remained significant. These five significant tests are linked to two loci. At the very least, the authors should re-run their analyses without locus Eaco_054 for which 43% of sites (3 of 7) are out of HWE using these methods of correction.

2. The authors ran Micro-checker, but the only results were reported in a single, unclear sentence (Line 200-201). What were the results? Which loci had evidence of null alleles or scoring errors? Do any issues arise in loci Eaco_001 or Eaco_054 where there is evidence of deviation from HWE?

3. The authors use a post-hoc AMOVA which is really somewhat superfluous. AMOVAs are designed to be used to test a priori hypotheses. Running an AMOVA using clusters that have already been determined using other analysis methods is redundant. I suggest that the authors remove their post-hoc AMOVA and instead, run a locus-by-locus AMOVA to see if whether there are any loci driving the genetic structuring they have measured between sites. If they find that a particular locus is driving all of the between-site differences, that locus will need to be evaluated for robustness.

4. Report the total number of alleles found at each locus. Having this data is helpful in evaluating the validity of loci. Only loci with shared alleles are useful in population genetic analyses.

The other major issue I found is with the IBD analysis. Correlations between geographic distance and genetic differentiation will be highly biased by hierarchical genetic structuring (see Meirmans 2012- The trouble with isolation by distance). With their neighbor-joining and Structure analyses, the authors have detected the kind of genetic clustering that will bias an IDB analysis. Given the scale of its genetic and geographic distance from all other sites, the authors should remove Nakuri from any IBD analysis (I suspect that removes the 6 most upper-right points). In addition, it would be a very good idea to distinguish the points that represent comparisons between sites in the Java Sea belonging to different Structure clusters (see my comment on Figure 2). If the correlation is being driven by these points, then there is actually no evidence for IDB, only genetic clustering.

Validity of the findings

The validity of the author’s findings and restoration/management implications are not fully supported until they address the issues I have discussed previously (particularly the issues with their loci). The author’s hypotheses for their results are heavily dependent on currents and oceanography, which are only truly examined using neutral genetic markers. While I suspect there will be no change in the genetic differentiation of Nakuri from the other sites, whether the genetic structuring they have found in the sites in the Java Sea proves robust and whether it is appropriate to conclude that it is likely current-influenced instead of a sign of local adaptation is dependent on further analysis of their loci.

Other notes on the Discussion:

Lines 256-260 – variance in genetic diversity is also highly effected by sample size

Lines 326-327 – what is this? There is nothing in the results as to how KJ and ANS show an inconsistent correlation between genetic/geographic distance. Provide more detail and explain.

---

## Round 0.2 · Minor Revisions

Overall both referees agree that the paper is much improved, but each also point out a number of minor edits that need to be made to the manuscript prior to publication. These are relatively minor, and I expect you should be able to address all these comments without difficulty. I look forward to seeing your revised manuscript.

Reviewer 1 ·

Basic reporting

The basic reporting has been much improved and the authors have addressed all previous comments. However, there are still some shortcomings that the authors need to address before the MS can be published.

Experimental design

improved.

Validity of the findings

The findings are valid.

Additional comments

L51: coalesced
L55: and frigate tuna
L62: need a reference for currents as provide reference for geological history. On that note, evolutionary, rather than geological, is correct
L62 – 71: this paragraph should be written slightly. It deals with both geological/evolutionary history of a region, but also the oceanography. To me it makes sense that historical effects are listed first and then maintained by contemporary processes. It would make it clearer if you wrote this paragraph in that order.
L90: and the rhizomes
L92: 10.2 days?
L93:the distances are unclear
L97: I disagree that the term ‘philopatric’ is appropriate for seagrasses. The term is more commonly used for animals and suggests a choice of staying or returning to a particular habitat, which seagrass seeds or suckers do not have.
L104: delete of
L111” what is ‘genet’?
L147: Micro-checker does not have a hyphen
L246: there is a whole section on genetic diversity here, and you mention that one of the major aims of the paper is to describe patterns of genetic diversity in this species. However, in the introduction the focus is on the historical and contemporary biogeography of this species without a mention of genetic diversity. This needs to be rectified. Ideally you would place this into a broader context of why genetic diversity is important, particularly in species that potentially have low dispersal (even though it seems from your study that even distantly situated populations are connected), i.e. reduced gen div at edge populations, more inbreeding etc. This is an important aspect.
L256: you show that genetic diversity is similar for your study to previous ones – what does that mean for this species? Good? Not so good?
L266: you need to build on this last sentence. At present this section is very descriptive and comparative. See my previous comments on why you need to investigate and better integrate a discussion on gen div.
L271 onwards: this is very confusing. You say that there is high structure, but then previously in L221 you say that samples separated by over 1000km are closely related? Which is it? Is the relationship between KJ and ANS different and why?
L365: lost
L367: above average is relative. Relative to what?
L376: there is a word missing here
L379: I think what is worth mentioning here that despite the loss of seagrass habitat, these plants manage to maintain good genetic diversity and heterozygosity. Should be noted.


References: there are numerous editorial issues with the references. Please ensure that you check these carefully. An example is L564: Molecular Ecology

Reviewer 2 ·

Basic reporting

The English is much improved with only a few minor errors which have been noted by line in the GENERAL COMMENTS section. The introduction provides good context for their particular study region and species.

Experimental design

The research is meaningful, methods and experimental design are clear.

Validity of the findings

The authors have clearly put a lot of effort into the improvement of this manuscript. There are three issues that need to be addressed in this version. These issues can be fixed with relatively small additions or modifications to their analyses and discussion.

1. IBD. The authors are moving in the right direction with the changes to their IDB analysis. However, their conclusions regarding the results remain wrong. They consistently state in the results, discussion, conclusion that there is significant IBD among all sites in E. acoroides. However, the data they present indicate that there is no IDB, only distinct hierarchical clustering. If one lumps two (or three) geographically distinct genetic clusters into one group and runs an IBD analysis as the authors have, the clustering will bias the results (creating a false positive). If the authors wish to run two or three of their clusters together in an IBD analysis, they need to use something like a partial mantel and test the correlation of genetic and geographic distance against a z-matrix of pairwise cluster associations (in other words, a third matrix of pairwise relationships where pairs are coded as either belonging to the same cluster, 0, or different clusters, 1). See Meirmans et al. 2012 for discussion of interpreting IDB results when hierarchical clustering is present. I don’t believe IBDWS is capable of running a partial mantel, however, there are several r packages that can.

The mantel test within cluster 2 is properly run and interpreted.

2. Issues with Eaco_054. In their response to previous review, the authors state that they reran their Fst analysis excluding Eaco_054 and they still generated significant p-values. I suggest they report that they did this in their manuscript and show the Fst table for the analysis without Eaco_054 with FDR corrected p-values in their supplemental material to support their results. The reason I suggest this is two-fold. 1) this locus deviates from HWE after FDR correction in 3 out of the 7 locations and 2) the total number of alleles at this locus is 29, which is more than the sample sizes at 3 locations and only 1 less than the sample sizes at 2 other locations. By this account, Eaco_054 is too polymorphic relative to sample size to be an ideal locus for population analysis. I feel it is important to show within their manuscript that its presence or absence makes no significant difference to their results.

3. The paragraph in lines 257-268 should be revised. Observed heterozygosity is LARGELY impacted by the researcher’s selection of markers. For example, here, the authors threw out a marker with Ho = 0 (Eaco_052) and do not report it in their estimates of Ho. They are correct to do so since it will be non-informative in population structure analysis, but it still exists. If the authors wish to discuss comparisons of Ho, they need to stick with comparisons between species or localities that have been analyzed with the same msat loci that they used.

Additional comments

Clarity:

Line 75 – This sentence comes across unclearly. I suggest changing it to something like, “These studies have found concordant phylogeographic breaks between populations in the Indian Ocean and Java Sea.“


Minor grammatical issues:

Line 51 – coalesce to coalesced

Line 58 – mangroves to mangrove

Line 64 – barrier to barriers

Line 75 – in Indian to in the Indian

Line 90 – rolled, the to rolled, and the

Line 93 – 0.1 63.5 km to 0.1-63.5 km

Line 102 – marker to markers

Line 104 – pattern to patterns

Line 148 – and 95% a confidence level to a 95% confidence level

Line 163 – population to populations

Line 171 – 100,000 steps of Markov Chain Monte Carlo (MCMC). to 100,000 steps of Markov Chain Monte Carlo (MCMC) sampling.

Line 197 – After correction for multiple test, to After correction for multiple tests,

Line 356 – becoming loss to declining (or something similar)

Line 391 – result also indicated to results also indicate

---

## Round 0.3 · Minor Revisions

The referee is happy with the revisions and is commented that the changes are both substantial and appreciated in terms of making the data accessible and analyses transparent. They also pointed out a number of grammatical errors that remain to be corrected, and are happy to see the paper published after these changes are made. If you need help with the grammar or additional edits, please let me know.

Reviewer 2 ·

Basic reporting

Good. The major issue with this version is grammatical errors in newly added sections. See Comments to the author.

Experimental design

no comment

Validity of the findings

no comment

Additional comments

Broad comments:

Fix your subsection numbering throughout the manuscript. (For example, Lines 116-117. The section number is 3, but the subsections are numbered 2.1, 2.2 etc.)

Check for consistency in the term isolation-by-distance across the manuscript. There are instances of “isolation by distance,” “isolation-by-distance,” and “Isolation By Distance.” I recommend changing them all to “isolation-by-distance.”

The authors have made several changes to their discussion. I found that these changes really strengthened the discussion of their results, however, it appears that the new discussion sections have the same grammatical issues that were present in earlier versions of the manuscript, and at this point fixing those is the largest thing that needs to be addressed in this manuscript. I spent time going through each section making comments and suggestions to assist with this.

In addition, if the authors use a location name in the manuscript, that location should be labeled in one of their figures. For example, Kalimantan is used in the discussion section, but the island is labeled only Borneo in the figures. Other locations discussed but not specifically labeled also include the Malaka Strait.

Grammar or other issues by section:

Abstract

LINE 26: remove comma after “taxon”

Introduction

LINE 42: remove comma before “seemingly” and remove comma after “Although”

LINE 45: change “genera” to “genus” and remove comma before “and”

LINE 57: change “population structures” to “population structure”

LINE 58: I recommend changing “simple” to “homogenous”

LINE 61: change “Euthynus” to “Euthynnus”

LINE 104: change “continue” to “continual”

LINE 113-114: “and infer on the effect of the Sunda Shelf and ….” is awkward. I recommend changing to something like “and to infer how the Sunda Shelf and regional currents shape the genetic patterns found in this species”

Materials and Methods

LINE 179: change “used Mantel test” to “used a Mantel test”

LINE 187-188: change “Both Mantel and partial Mantel test was performed” to “Both Mantel and partial mantel tests were performed”

Results

LINE 196: change “the Eaco_050” to “Eaco_050”

LINE 209-211: change “revealed five loci (…) that deviated from” to “revealed that five loci (…) deviated from”

LINE 212: change “prior to multiple test” to “prior to a multiple test”

LINE 223: change “FST test” to “FST tests”

LINE 224-226: I recommend changing “; our findings were not different between datasets (…) Therefore, all loci” to something like “Removing Eaco_054 did not change the results of pairwise comparisons (…), therefore all six loci were included in further analyses. ”

In addition to the above comment, I noticed that Supplemental Table S1 was the raw allelic data for each individual, not the results of tests of pairwise differentiation. So I recommend removing the reference in the above sentence and adding a sentence at the end of this paragraph along the lines of: “Individual genotypes for these loci are reported in Supplementary material, Table S1.”

LINE 230: change “highly significant” to “significant” (with p-values, results are either significant or not, there are no levels of significance).

LINES 231-232: change “highest” and “lowest” to “largest” and “smallest”

LINES 246-247: I recommend changing “After performing partial Mantel test, the IBD result still remained significant with P=0.03 (Fig.3).” to “A partial Mantel performed on the remaining Java and Natuna Sea locations indicated that IBD is present across these sites (p=0.03, Fig. 3).”

Discussion

LINE 251: change “number of alleles) genetic” to “number of alleles), genetic”

LINE 266: remove the comma before “and”

LINE 269: change “heterozigosity” to “heterozygosity”

LINE 270: can this personal observation be attributed to a specific author or authors on this manuscript? If so, I recommend using “ (pers. comm. Author Name(s)).”

LINE 271: change “area are tend” to “areas tend”

LINE 272: change “heterozigosity” to “heterozygosity”

LINE 271-273: The sentence “Although ANS… undisturbed area.” is awkward. I recommend changing to something like “Although ANS had the lowest observed heterozygosity (Ho), sampling at this location was also conducted in an undisturbed area so the lower Ho may be the result of other factors.”

LINE 273-274: This sentence is awkward. I recommend changing to something like “Microchecker results indicated that ANS had an excess of homozygosity, a potential indicator of inbreeding.”

LINE 275: change “occur” to “occurs”

LINE 275: change “reduce” to “reduces”

LINE 286: change “in Indo Malay Archipelago” to “in the Indo-Malay Archipelago”

LINE 289: remove comma before “and”

LINE 300: change “a result of” to “due to” so that the word result is not used twice in this sentence

LINE 310: change “seed” to “seeds” and “does” to “do”

LINES 315-316: change “Although, clustering output also showed that there were any secondary peak” to “Clustering output indicated secondary peaks”

LINE 316: I recommend changing “This results implied…major groups.” to “These results indicate the presence of further substructure similar to that found in FST analysis.”

LINE 318: change “TD and PR. While at K=7” to “TD and PR, while at K=7”

LINE 332: change “Euthynnus affin” to “Euthynnus affinis”

LINE 337: change “scale” to “scales”

LINE 341-342: change “this study, mirrors the genetic structure of genus” to “this study mirrors the genetic structure found in the genus”

LINES 345-346: change “found an appropriate habitat across Seribu Islands, Bangka, Batam and used it as stepped stone to a long distance dispersal.” to “found appropriate habitat across the Seribu, Bankga, and Batam Islands and used these islands as stepping stones for long distance dispersal.”

LINE 353: change “across Java” to “across the Java”

LINE 357: change “result” to “results”

LINE 359: change “find” to “found”

LINES 360-361: change “This genetic pattern probably influenced by sea level changes during Pleistocene” to “This genetic pattern may also have been influenced by sea level changes during the Pleistocene”

LINE 363: change “seemed” to “seem”

LINE 364: change “this connection could be facilitated the gene flow” to “this connection may have facilitated gene flow”

LINE 365-366: change “creating” to “create” and “maintaining” to “maintain”

LINE 367: change “maintenance” to “maintain”

PARAGRAPH 359-370: This paragraph is a very good inclusion but I feel like the last two sentences need to be deleted. They are not specific to the paragraph topic and the citations used are found in other places in the manuscript. The beginning of the paragraph is good – a shifting coastline that at one point included two contemporary (and distant) island habitats while sea levels rose and the Sunda Shelf was flooded is a good hypothesis for the genetic patterns the authors found in the Natuna and Java Seas. However, long distance dispersal via hurricanes or typhoons that might have specifically affected only those two islands and not the other sites sampled in this region is not helpful or a particularly strong explanation for the author’s findings. The beginning of this paragraph could possibly be combined with the following paragraph.

LINE 371: change “Seagrasses population in Natuna seas could be originating from any different source.” to “Seagrass populations in the Natuna Sea could be originating from a different source.”

LINE 372: change “area in Natuna Sea is supplied by a larval transport from South China” to “areas in the Natuna Sea are supplied by larval transport from the South China”

LINES 373-374: change “Although, others study showed that population in Natuna might also come from the Andaman Sea via Malaka strait following the ice retreat” to “Other studies showed that populations in the Natuna Sea may also receive propagules from the Andaman Sea via the Malaka Strait following the ice retreat”


LINE 375: change “population” to “populations” and “guess” to “suspect”

LINE 376: change “barrier” to “barriers”

LINE 377: change “part of Natuna Sea” to “parts of the Natuna Sea”

LINE 378-379: change “to reveal the origin ” to “to better understand the origins of the populations”

LINE 379: change “Malaka strait and South China Sea” to “the Malaka Strait and the South China Sea”

LINE 383: change “organism” to organisms”

LINE 384: change “ecosystem” to “ecosystems”

LINE 386: change “ have been conducted” to “is underway”

LINE 391: change “Structure” to “Bayesian”

LINE 391: change “recommend at least there are three” to “recommend resource managers treat populations across our sampling region as three”

LINE 392: change “to maintain unique genetic characteristics of the region for seagrasses conservation” to “to maintain the unique genetic characteristics of each region to support seagrass conservation”

PARAGRAPHS 389-398 and 399-405: The information in these two paragraphs should be combined into a single paragraph. There is a lot of redundancy between them.

PARAGRAPH 406-415: I recommend the first sentence in this paragraph be moved to an appropriate place in the proceeding paragraph(s), and the last part of this paragraph should be moved down to the Conclusions section.

Conclusion

LINE 418: change “heterozygosities between” to “heterozygosity in”

---

## Round 0.4 · accepted · Accept

Thanks for your revisions, and for selecting PeerJ for your paper. I am happy to move your submission forward into production.

Aloha,
Rob